# Fully Automated Regional Analysis of Myocardial T2* Values for Iron Quantification Using Deep Learning

**Nicola Martini** [1,2,3,†] , **Antonella Meloni** [1,2,†] , **Vincenzo Positano** [1,2] , **Daniele Della Latta** [2,3] , **Petra Keilberg** [1] , **Laura Pistoia** [1] , **Anna Spasiano** [4] , **Tommaso Casini** [5] , **Angelica Barone** [6] , **Antonella Massa** [7] , **Andrea Ripoli** [2,3] **and Filippo Cademartiri** [1,*]

1   Department of Radiology, Fondazione G. Monasterio CNR-Regione Toscana, 56124 Pisa, Italy
2   Unità Operativa Complessa di Bioingegneria, Fondazione G. Monasterio CNR-Regione Toscana, 56124 Pisa, Italy
3   Deep Health Unit, Fondazione G. Monasterio CNR-Regione Toscana, 56124 Pisa, Italy
4   Unità Operativa Semplice Dipartimentale Malattie Rare del Globulo Rosso, Azienda Ospedaliera di Rilievo Nazionale "A. Cardarelli", 80131 Napoli, Italy
5   Centro Talassemie ed Emoglobinopatie, Ospedale "Meyer", 50139 Firenze, Italy
6   Unità Operativa di Pediatria e Oncoematologia, Dipartimento Materno-Infantile, Azienda Ospedaliero-Universitaria di Parma, 43100 Parma, Italy
7   Servizio Trasfusionale, Ospedale "Giovanni Paolo II", 07026 Olbia, Italy
*   Correspondence: fcademartiri@ftgm.it; Tel.: +39-050-3152817
†   These authors contributed equally to this work.

**Abstract:** Cardiovascular magnetic resonance (CMR) T2* mapping is the gold standard technique for the assessment of iron overload in the heart. The quantitative analysis of T2* values requires the manual segmentation of T2* images, which is a time-consuming and operator-dependent procedure. This study describes a fully-automated method for the regional analysis of myocardial T2* distribution using a deep convolutional neural network (CNN). A CNN with U-Net architecture was trained to segment multi-echo T2*-weighted images in 16 sectors in accordance with the American Heart Association (AHA) model. We used images from 210 patients (three slices, 10 multi-echo images) with iron overload diseases to train and test the CNN. The performance of the proposed method was quantitatively evaluated on an independent holdout test set by comparing the segmentation accuracy of the CNN and the T2* values obtained by the automated method against ground-truth labels provided by two experts. Segmentation metrics and global and regional T2* values assessed by the proposed DL method closely matched those obtained by experts with excellent intraclass correlation in all myocardial sectors of the AHA model (ICC range [0.944, 0.996]). This method could be effectively adopted in the clinical setting for fast and accurate analysis of myocardial T2*.

**Keywords:** T2* mapping; cardiovascular magnetic resonance; segmentation; deep learning; myocardial iron overload; quantitative MRI

## 1. Introduction

Iron overload, designated as excess stores of iron in the body, is a surprisingly common condition affecting millions of people worldwide. It can result from inherited disorders of iron homeostasis characterized by increased intestinal iron absorption (primary hemochromatosis) and from chronic transfusions in patients with acquired and inherited anemias such as thalassemia and sickle cell disease (secondary hemochromatosis) [1]. Excess iron is deposited in the parenchyma of many tissues and, being toxic, can cause tissue damage and organ dysfunction, especially in the heart, liver, and pancreas [2]. In particular, iron-induced cardiomyopathy continues to be a leading cause of morbidity and mortality in patients with primary as well as secondary hemochromatosis [3]. If diagnosed and treated in its early stages, this condition is treatable and reversible [4]. Therefore, the quantification of myocardial iron overload (MIO) is the key to better patient management.

To date, cardiovascular magnetic resonance (CMR) is the only technique able to provide a quantitative assessment of the cardiac iron burden in a non-invasive manner [5]. The presence of myocardial iron deposits causes microscopic magnetic field inhomogeneities and results in a reduction in all relaxation times (T1, T2, and T2*) [6], with the T2* technique representing the gold standard. The T2* value is calculated by fitting the CMR signal acquired at different echo times (TEs) with a proper exponential decay model [7]. Two validated approaches are generally used in the clinical setting: the single-slice and the multislice approach. In the single-slice approach, the T2* value is calculated in a single region of interest (ROI) delineated in the mid-ventricular septum [8], while the multislice approach has been developed for the regional analysis of the entire left ventricle (LV) [9]. The multislice approach allows the identification of an early and heterogeneous iron distribution that would remain otherwise undiscovered with a single measurement in the mid-ventricular septum [10], but it is more tedious, time-consuming, and operator-dependent. In fact, users are required to manually delineate the LV boundaries and identify the right ventricle (RV) insertion point to divide the LV into equiangular sectors, in accordance with the American Heart Association (AHA) model [11].

Deep learning (DL), a branch of artificial intelligence (AI), is well-suited to streamline this workflow, offering the promise of full automation and output consistency and repeatability. DL algorithms learn efficient features or patterns directly from the input data (raw data) and combine these features for classification without the need for prior knowledge and human intervention [12].

The most successful type of DL algorithms for image analysis are the convolutional neural networks (CNNs) [13]. The architecture of a typical CNN network is composed of multiple layers ("hidden layers"), such as convolution layers, pooling layers, and fully connected layers that, in turn, map the input image to the desired output while learning increasingly higher-level imaging features through a backpropagation algorithm [14]. Convolution layers are the core building blocks of the CNN that perform feature extraction by applying a set of filters to the image to produce spatially dependent features. Pooling layers perform downsampling operations to decrease the spatial size of the representation and the amount of computation and weights. In classification tasks, the extracted features are finally mapped by fully connected layers to the final outputs, such as the probabilities for each class [15]. Instead, in semantic segmentation tasks, fully convolutional networks (FCN) trained end-to-end, pixels-to-pixels, have shown to obtain the best results [16].

In the last few years, CNNs have made a tremendous impact on the entire workflow in CMR imaging [17], bringing unprecedented benefits in the areas of image acquisition [18], image reconstruction [19,20], image segmentation [21], and diagnostic evaluation [22]. It has been shown that CNNs can be used to process multiple CMR sequences (cine, late gadolinium enhancement, native and post-contrast T1), achieving robust segmentation results and leading to quantitative results (i.e., ejection fractions or T1 values) close to those obtained with manual segmentation [23–25].

To the best of our knowledge, no deep learning–based analysis platform has yet been developed for automating the global and regional analysis of myocardial T2* values. In this work, we sought to develop and validate an automatic method for the AHA model segmentation of T2* mapping using a deep convolutional neural network. The performance of the proposed method was quantitatively evaluated on an independent holdout test set by comparing the segmentation accuracy of the CNN and the T2* values obtained by the automated method against the ground-truth contours and measurements provided by expert operators.

## 2. Materials and Methods

### 2.1. Study Population

Images from 210 patients with iron overload diseases (106 females, 38.2 ± 12.8 years) were retrospectively studied. All of the patients were consecutively enrolled from 2015 to 2018 in the coordinator center of the Extension-Myocardial Iron Overload in Thalassemia

(E-MIOT) project (Pisa). E-MIOT is an Italian network composed of 66 thalassemia centers and 11 magnetic resonance imaging (MRI) sites that perform CMR exams using homogeneous, standardized, and validated procedures [26,27].

The study complied with the Declaration of Helsinki and was approved by the institutional ethics committee. All patients provided written informed consent to the protocol.

### 2.2. Image Acquisition and T2* Map Generation

CMR was performed on two clinical 1.5T scanners (Signa CVi or Signa Artist, GE Healthcare, Milwaukee, WI, USA) using a cardiac phased-array receiver surface coil for signal reception.

Three slices (basal, medial, and apical) of the LV in short-axis view were collected using an ECG-triggered T2* gradient–echo multi-echo sequence [9]. The images were acquired during the tele-diastolic phase to minimize cardiac motion [28]. The multi-echo sequence parameters were as follows: number of echo times, 10; first echo time, 2.0 ms; echo spacing, 2.26 ms; flip angle, 25°; matrix, 192 × 256 pixels; field of view (FOV), 35 × 35 cm; bandwidth, 62.5 KHz; slice thickness, 8.0 mm; and views per segment, 6–8.

The fully automatic process for T2* map computation has already been described and validated [29]. Briefly, the T2* value for each pixel was calculated by fitting the corresponding MRI signal at the increasing TEs with a single exponential decay model:

$$S = S_0 e^{\frac{-TE}{T2*}} \tag{1}$$

where S indicates the signal intensity, $S_0$ is the signal intensity at TE = 0, and TE represents the echo times. The fitting was performed using the Levenberg–Marquardt algorithm. In patients with severe MIO, the signal decays quickly and becomes comparable to image noise, thus hampering the goodness and generating a high fitting error, computed as the normalized mean root square error between the measured MR signal and the fitted decay curve. If the fitting error was more than 5%, the algorithm discarded the last TE and performed the fitting again. The procedure was iterated until the error became <5% or the number of TEs became equal to three.

### 2.3. Data Preparation and Labeling

The acquired T2* images were analyzed by an expert CMR operator (P. K., >15 years of experience in T2* analysis) using a previously validated, custom-written IDL-based software (Hippo-MIOT®) to obtain ground-truth data [29]. Briefly, for each short-axis slice, the operator delineated the endocardial and epicardial borders of the LV in an image corresponding to the first or second echo time and defined a reference point in the anterior septal insertion of the right ventricle. The myocardium defined in the previous step was automatically segmented into equiangular segments starting from the reference point. According to the AHA model, the basal and medial slices were divided into 6 segments and the apical slice into 4 segments [11]. The T2* values in each segment was obtained by averaging the T2* values for all the pixels within the segment. The T2* values of each slice were obtained by averaging the T2* values of all myocardial pixels, regardless of the AHA sector. The ground-truth segmentation masks and T2* maps were stored in Hierarchical Data Format version 5 (HDF5) for DL model training and performance evaluation.

### 2.4. Deep Learning Segmentation

The segmentation of three slices into 16 AHA-based myocardial segments was translated into a single-slice multiclass semantic segmentation problem with the number of predicted classes equal to 7, i.e., the six myocardial segments and the background. To this end, the ground-truth masks of the apical slice were subdivided into 6 segments for the training, splitting the septal and lateral segments into two.

A fully convolutional neural network architecture (U-Net) [30] was adopted for segmentation. Our U-Net model was fed with full FOV single-slice raw T2* multi-echo images (array of 256 × 256 × 10) and was trained to give as output the corresponding 7-classes

mask (Figure 1). Each block of our U-Net consisted of two convolutional layers with a kernel size of $5 \times 5$ (stride 1 and padding 1) with a $1 \times 1$ convolutional layer in between to reduce the depth of the feature maps and, in turn, reduce the total number of network parameters. The Rectified Linear Unit (ReLU) was used as an activation function, and batch normalization layers were inserted after each convolutional layer. Max pooling layers were used for downsampling, while transposed convolution layers (stride 2, padding 2) performed the upsampling of the feature maps to restore the original size of the input image. The output of the last convolutional layer was a $256 \times 256 \times 7$ array of scores representing the 7 segmented classes, which were converted to normalized probability values via a softmax function. Finally, the predicted segmentation mask was obtained by pixel-wise application of the *argmax* function on softmax probabilities.

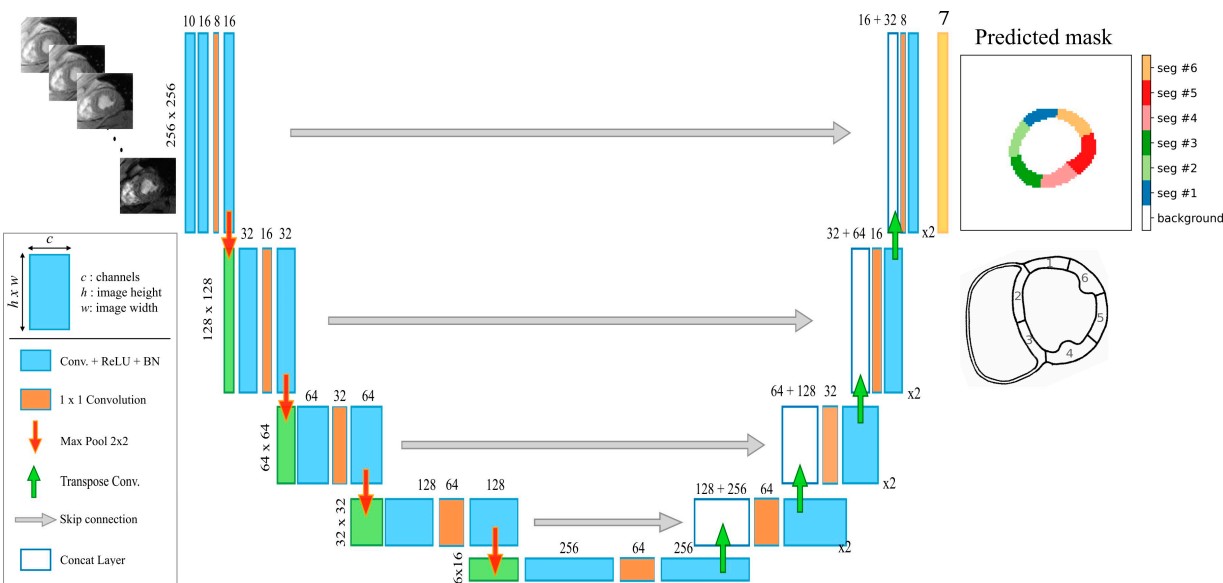

**Figure 1.** The proposed U-Net model architecture for the automatic segmentation of multi-echo gradient-echo images.

### 2.5. Network Training

The proposed U-Net model was developed in Python using the TensorFlow and Keras framework. Network training was performed on a computer with the following specifications: 12-core Intel i7-7800 CPU 3.5 GHz with 64 GB RAM and an NVIDIA Titan Xp GPU with 12 GB of memory, and running the Linux operating system.

The whole dataset of the T2* multi-echo images was split into a training set (N = 510 slices), validation set (N = 60 slices), and test set (N = 60 slices) for performance evaluation. During training, random rotations (range $[-90°, +90°]$) and scaling transformations (range [0.8, 1.2]) were used for data augmentation. A linear interpolation was used for transforming input images, while nearest-neighbor interpolation was applied to the ground-truth label masks.

Adam optimizer with initial learning rate of 0.001 ($\beta1 = 0.9$, $\beta2 = 0.999$, epsilon = 1e-7) was used with a batch size of 16. The weighted cross-entropy (WCE) loss was chosen as a loss function to address the issue of class imbalance. The class weights were set inversely proportional to the frequencies of each class label in the training data. Iterative training and validation steps were carried out on 2000 epochs, with 32 steps for each epoch, performing each validation at the end of each epoch. The final model was selected as the one achieving the best performance on the validation set.

### 2.6. Evaluation of the Model Performance

The performance of the automated method was evaluated in two ways: using commonly used metrics for the assessment of the segmentation accuracy and using quantitative measures (segmental and global T2* values) derived from the segmentations.

#### 2.6.1. Segmentation Accuracy Assessment

The Dice similarity coefficient, average perpendicular distance, and Hausdorff distance were used as metrics of segmentation performance.

The Dice similarity coefficient (DSC) evaluates the overlap between the automated or predicted segmentation mask A and the manual segmentation mask B, and it is defined as:

$$DSC = \frac{2|A \cap B|}{|A| + |B|} \tag{2}$$

DSC ranges between 0 and 1, with 0 denoting no overlap and 1 denoting perfect agreement. The higher the DSC metric, the better the agreement. Following previous studies, a DSC > 0.7 was defined as substantial agreement, between 0.4 and 0.7 as moderate agreement, and <0.4 as large variation [31,32].

The average perpendicular distance (APD) distance and Hausdorff distance (HD) evaluate, respectively, the mean and the maximum of the minimum surface distances between the segmentation contours $\partial A$ and $\partial B$. They are defined as:

$$APD = \frac{1}{2|\partial A|} \sum_{p \in \partial A} d(p, \partial B) + \frac{1}{2|\partial B|} \sum_{q \in \partial B} d(q, \partial A) \tag{3}$$

$$HD = max\left(max_{p \in \partial A} d(p, \partial B), max_{q \in \partial B} d(q, \partial A)\right) \tag{4}$$

where $d(p, \partial)$ denotes the minimal distance from point p to contour $\partial$. These distances indicate how much the contours should be quantitatively modified relative to the ground truth. The lower the distance metric, the better the agreement.

In addition to these commonly used segmentation metrics, we additionally proposed a custom metric called Angular Error (AE) for the assessment of the correct orientation of each myocardial segment. The AE was calculated as:

$$AE = cos^{-1}\left(\frac{b \cdot a}{\|b\|\|a\|}\right) = cos^{-1}\left(\frac{\vec{CB} \cdot \vec{CA}}{\|\vec{CB}\|\|\vec{CA}\|}\right) \tag{5}$$

where b and a denote the vectors between the centroids *B* and *A* of the predicted and the manual segmental masks, respectively, and the center of the myocardium (point *C*). The lower the AE metric, the better the agreement.

The DSC, APD, and HD metrics were calculated for each myocardial segment separately, for the entire myocardial wall in the three slices (in the following denoted by Myoc) and in the left ventricle cavities (denoted by LV).

#### 2.6.2. Accuracy of T2* Quantification

All of the data were analyzed and visualized using Python 3.6 packages (NumPy, SciPy, Pandas, Matplotlib, Seaborn). The continuous variables are reported as mean ± standard deviation (SD).

The correlation between the segmental and global T2* values obtained with the two different approaches (manual vs automatic segmentation) was evaluated with the Spearman test since the T2* values showed a non-normal distribution. The summary data were displayed using scatter plots with regression lines.

The coefficient of variation (CoV) was obtained as the ratio of the SD of the half mean square of the differences between the repeated values to the general mean. A CoV < 10% was considered good. The ICC was obtained from a two-way random effects model with measures of absolute agreement. An ICC $\geq$ 0.75 was considered excellent, between 0.40 and 0.75 fair to good, and <0.40 poor [33].

The agreement between the DL algorithm and the manual operator was determined by the Bland–Altman technique, plotting the difference versus the average of the variables. Bias was the mean of the difference between the two methods, and agreement was the mean $\pm$ 1.96 SD. In all of the tests, two-tailed $p < 0.05$ was considered statistically significant.

### 2.6.3. Assessment of Validity

The "conservative" value of 20 ms is commonly used as the lower limit of normal for the segmental and global heart T2* values at 1.5T [5].

The global T2* values obtained with manual segmentation were considered the standard of reference and were used to categorize patients into two clinically relevant groups: MIO (T2* < 20 ms) and no MIO (T2* $\geq$ 20 ms). So, it was detected if classifications obtained for fully-automated global heart T2* values were true positive (TP), false positive (FP), true negative (TN), and false negative (FN). The sensitivity, specificity, positive and negative predictive values (PPV and NPV), and accuracy were calculated.

### 2.6.4. Inter-Observer Agreement

The images of the test set were also analyzed by a second observer to evaluate the inter-observer agreement and to compare the inter-observer variability with the performance of the automatic method. The images were presented in random order to another operator (V.P., >15 years of experience), blinded to the results obtained by the first observer. The segmentation metrics (DSC, APD, HD, and AE) and quantitative T2* values were evaluated between the pair of observers and compared against the DL method.

## 3. Results

*Performance of U-Net*

The proposed deep learning method was able to provide labels for all the 320 myocardial segments of the test data (20 patients $\times$ 16 segments). Figure 2 illustrates the predicted segmentation of the LV according to the 16-segment AHA model. It shows that the automated segmentation agrees well with the manual segmentation of the clinical expert.

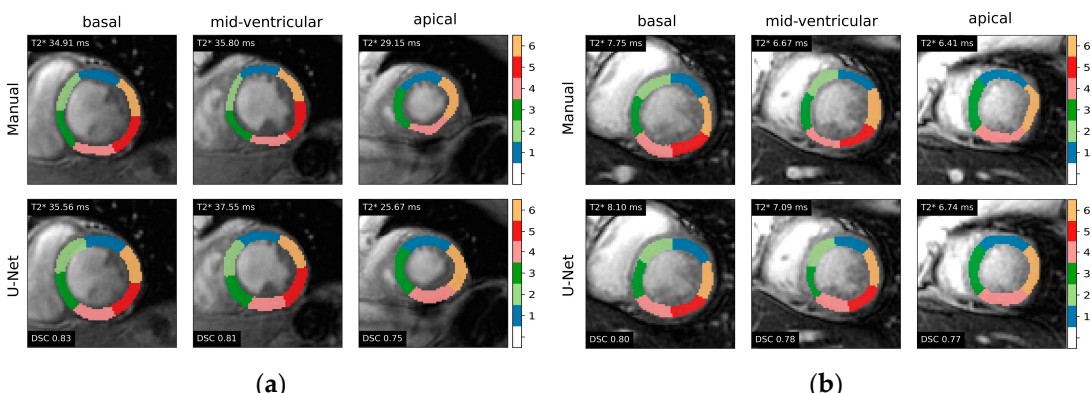

**Figure 2.** Representative examples of 16-segment AHA model myocardial segmentation from gradient-echo multi-echo T2* images in two testing cases: (**a**) subject with normal T2* values and (**b**) patient with iron overload. Ground-truth label masks (upper panels) and U-Net-based segmentations (lower panels) are shown for each of the three myocardial slices. Global T2* derived from the segmentation and Dice similarity coefficients (DSC) are also indicated.

The segmentation metrics evaluated in the test set (Table 1) demonstrated excellent agreement between the manual and fully-automated segmentations obtained by the U-Net in terms of volume overlap (mean Dice 0.771 [CI 0.755–0.787]), surface distance (APD = 0.61 mm [CI 0.53–0.69], HD = 4.30 mm [CI 3.95–4.65]), and angulation error (AE = 3.97° [CI 3.48–4.47]) across all segments.

**Table 1.** Dice similarity coefficients (DSC), average perpendicular distance (APD), Hausdorff distance (HD), and angulation error (AE) between automated and manual segmentations calculated for each of the 16 sectors of the AHA model, for the myocardial wall (Myoc) and for LV cavity (LV).

| Segment/Myoc/LV | DSC | APD [mm] | HD [mm] | AE [°] |
|---|---|---|---|---|
| 1: basal anterior | $0.795 \pm 0.086$ | $0.42 \pm 0.29$ | $3.25 \pm 1.60$ | $3.44 \pm 3.78$ |
| 2: basal anteroseptal | $0.794 \pm 0.081$ | $0.41 \pm 0.26$ | $3.68 \pm 1.72$ | $2.81 \pm 3.22$ |
| 3: basal inferoseptal | $0.805 \pm 0.084$ | $0.37 \pm 0.24$ | $3.31 \pm 1.86$ | $2.98 \pm 3.17$ |
| 4: basal inferior | $0.807 \pm 0.101$ | $0.38 \pm 0.31$ | $3.11 \pm 1.34$ | $2.36 \pm 2.59$ |
| 5: basal inferolateral | $0.767 \pm 0.191$ | $0.57 \pm 0.88$ | $3.44 \pm 2.28$ | $3.90 \pm 3.46$ |
| 6: basal anterolateral | $0.741 \pm 0.186$ | $0.63 \pm 0.76$ | $3.39 \pm 1.93$ | $3.27 \pm 2.57$ |
| 7: medium anterior | $0.746 \pm 0.157$ | $0.57 \pm 0.52$ | $3.87 \pm 1.86$ | $3.87 \pm 4.91$ |
| 8: medium anteroseptal | $0.772 \pm 0.096$ | $0.44 \pm 0.28$ | $3.39 \pm 1.06$ | $3.49 \pm 3.19$ |
| 9: medium inferoseptal | $0.787 \pm 0.145$ | $0.44 \pm 0.52$ | $3.32 \pm 1.45$ | $3.31 \pm 3.31$ |
| 10: medium inferior | $0.764 \pm 0.146$ | $0.53 \pm 0.53$ | $3.58 \pm 1.77$ | $3.45 \pm 4.51$ |
| 11: medium inferolateral | $0.752 \pm 0.160$ | $0.61 \pm 0.70$ | $4.04 \pm 2.32$ | $3.27 \pm 4.94$ |
| 12: medium anterolateral | $0.754 \pm 0.203$ | $0.64 \pm 0.93$ | $3.76 \pm 2.38$ | $4.23 \pm 5.39$ |
| 13: apical anterior | $0.762 \pm 0.129$ | $0.54 \pm 0.48$ | $4.09 \pm 2.05$ | $5.84 \pm 5.65$ |
| 14: apical septal | $0.775 \pm 0.096$ | $0.47 \pm 0.35$ | $4.06 \pm 1.73$ | $5.01 \pm 5.78$ |
| 15: apical inferior | $0.762 \pm 0.133$ | $0.53 \pm 0.50$ | $4.11 \pm 2.11$ | $5.56 \pm 5.48$ |
| 16: apical lateral | $0.730 \pm 0.168$ | $0.68 \pm 0.68$ | $4.43 \pm 2.43$ | $7.08 \pm 6.12$ |
| Myoc: basal | $0.819 \pm 0.087$ | $1.94 \pm 0.92$ | $13.18 \pm 4.78$ | $3.13 \pm 2.27$ |
| Myoc: mid-ventricular | $0.802 \pm 0.095$ | $2.11 \pm 0.96$ | $13.31 \pm 5.81$ | $3.60 \pm 3.77$ |
| Myoc: apical | $0.802 \pm 0.090$ | $2.12 \pm 1.00$ | $13.58 \pm 6.51$ | $5.87 \pm 4.35$ |
| LV: basal | $0.927 \pm 0.032$ | $2.44 \pm 1.73$ | $13.45 \pm 8.39$ | n.a. |
| LV: mid-ventricular | $0.907 \pm 0.044$ | $3.06 \pm 2.01$ | $16.32 \pm 7.32$ | n.a. |
| LV: apical | $0.911 \pm 0.043$ | $2.24 \pm 1.29$ | $13.23 \pm 5.77$ | n.a |

In the whole myocardial wall, the Dice values of the three slices were all above 0.8. The Dice and angulation error metrics showed better values for basal than mid-ventricular and apical slices, with the AE being significantly higher in the apical slice (basal vs apical: $3.13 \pm 2.33$ vs. $5.87 \pm 4.46$, $p < 0.0001$; mid-ventricular vs. apical: $3.60 \pm 3.87$ vs. $5.87 \pm 4.46$, $p < 0.05$).

In the LV cavity, the Dice values of the three slices were all above 0.9, and the APD values were all less than 3 mm.

In the test cohort, a strong correlation between the segmental and global T2* values obtained with the manual and the DL-based methods was obtained (Table 2). The CoV was always <10%, and the ICC was excellent.

**Table 2.** Agreement between T2* values obtained with the manual and the DL-based methods in the test cohort.

| Segment | Bland Altman Bias (Limits of Agreement) [ms] | Correlation (R; *p*-Value) | ICC (95%CI) | CoV (%) |
|---|---|---|---|---|
| 1: basal anterior | −0.97 (−5.15 to 3.1) | R = 0.955; *p* < 0.0001 | 0.994 (0.983–0.998) | 5.42 |
| 2: basal anteroseptal | −0.38 (−4.13 to 3.37) | R = 0.977; *p* < 0.0001 | 0.996 (0.990–0.998) | 4.12 |
| 3: basal inferoseptal | 0.09 (−3.64 to 3.83) | R = 0.982; *p* < 0.0001 | 0.996 (0.990–0.998) | 3.97 |
| 4: basal inferior | 0.10 (−2.63 to 2.83) | R = 0.992; *p* < 0.0001 | 0.997 (0.993–0.999) | 3.42 |
| 5: basal inferolateral | 1.11 (−5.63 to 7.85) | R = 0.919; *p* < 0.0001 | 0.971 (0.928–0.989) | 9.99 |
| 6: basal anterolateral | −0.09 (−4.05 to 3.86) | R = 0.953; *p* < 0.0001 | 0.994 (0.985–0.998) | 4.56 |
| 7: medium anterior | −1.09 (−7.19 to 5.01) | R = 0.962; *p* < 0.0001 | 0.987 (0.966–0.995) | 8.37 |
| 8: medium anteroseptal | −0.64 (−4.45 to 3.17) | R = 0.992; *p* < 0.0001 | 0.995 (0.987–0.998) | 4.67 |
| 9: medium inferoseptal | −0.97 (−4.28 to 2.35) | R = 0.995; *p* < 0.0001 | 0.996 (0.986–0.999) | 4.17 |
| 10: medium inferior | −1.46 (−5.57 to 2.63) | R = 0.988; *p* < 0.0001 | 0.992 (0.964–0.997) | 6.77 |
| 11: medium inferolateral | −0.74 (−6.02 to 4.55) | R = 0.961; *p* < 0.0001 | 0.986 (0.966–0.995) | 7.64 |
| 12: medium anterolateral | −1.03 (−5.54 to 3.48) | R = 0.979; *p* < 0.0001 | 0.990 (0.974–0.996) | 6.32 |
| 13: apical anterior | −0.08 (−5.37 to 5.21) | R = 0.982; *p* < 0.0001 | 0.992 (0.979–0.997) | 6.71 |
| 14: apical septal | −0.41 (−3.68 to 2.86) | R = 0.991; *p* < 0.0001 | 0.997 (0.993–0.999) | 3.87 |
| 15: apical inferior | −0.09 (−5.26 to 5.07) | R = 0.991; *p* < 0.0001 | 0.993 (0.982–0.997) | 6.61 |
| 16: apical lateral | 0.92 (−4.65 to 6.49) | R = 0.961; *p* < 0.0001 | 0.988 (0.969–0.995) | 6.93 |
| Global | −0.36 (−2.14 to 1.42) | R = 0.992; *p* < 0.0001 | 0.998 (0.996–0.999) | 0.233 |

There was a strong correlation between the automated and manual T2* measurements in the per-slice (Figure 3a) and per-segment (all together; Figure 3c) analyses. The Bland–Altman plot (Figure 3b,d) showed narrow limits of agreement (per-slice: 3 ms; per-sector: 5 ms) between the two assessments with no bias (<1 ms).

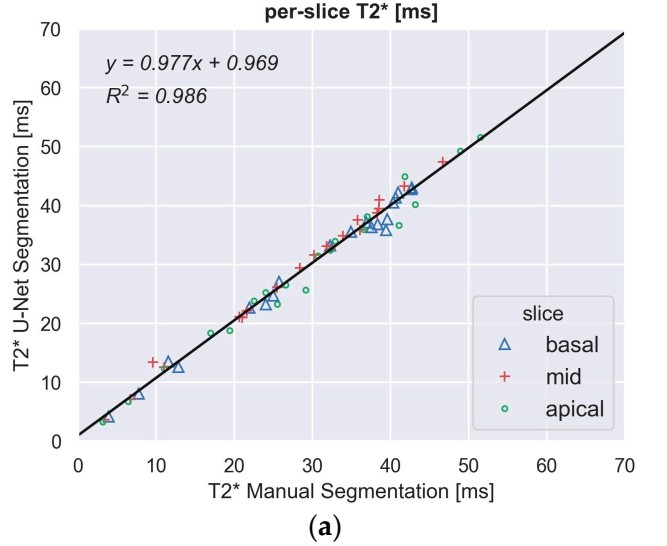
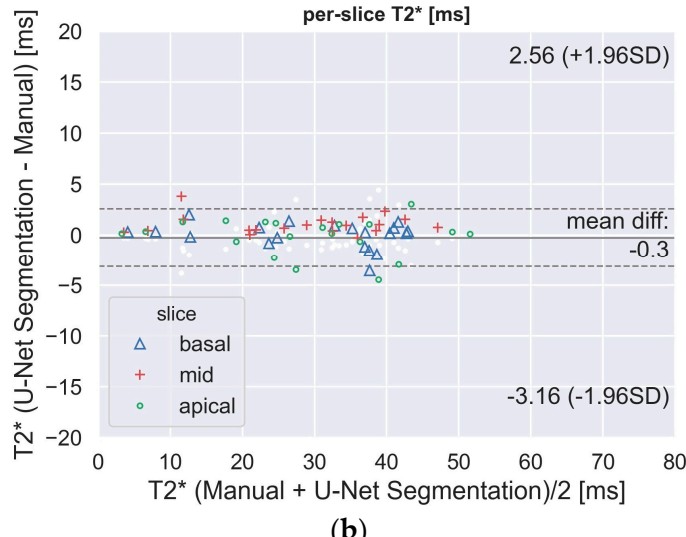

**Figure 3.** *Cont.*

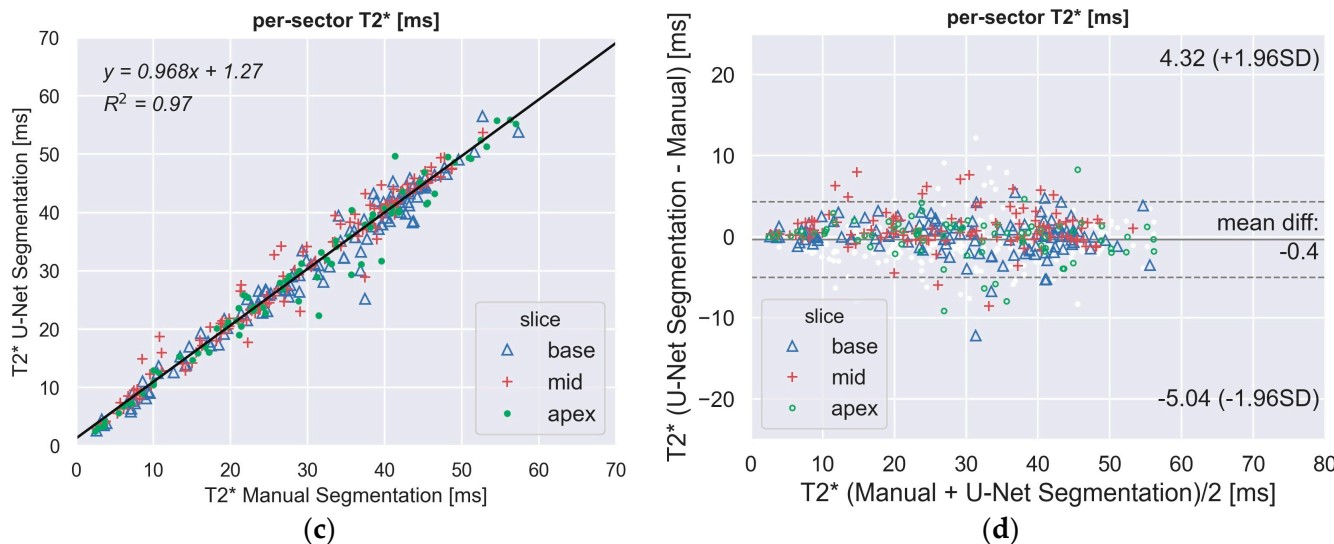

**Figure 3.** (**a**,**c**) Scatter plots demonstrating a high correlation between fully-automated DL method and manually processed segmental T2* values. The segments of the three slices (basal, mid-ventricular, apical) are indicated by different markers. (**b**,**d**) Bland–Altman plot showing the excellent agreement between the manual and the proposed DL method for the regional quantitative analysis of T2*. Middle line denotes mean. Dashed lines denote ± 1.96 standard deviations.

Table 3 shows the efficacy of the U-NET in differentiating the normal from pathological T2* values. The sensitivity was under 80% for three myocardial segments, while specificity was always very strong (high 90's). The accuracy was very high (high 90's). The global heart T2* value was always correctly classified.

**Table 3.** Performance evaluation of automated identification of patients with reduced T2*.

| | Sensitivity (%) | Specificity (%) | Positive Predictive Value (%) | Negative Predictive Value (%) | Accuracy (%) |
|---|---|---|---|---|---|
| 1: basal anterior | 100 | 100 | 100 | 100 | 100 |
| 2: basal anteroseptal | 100 | 100 | 100 | 100 | 100 |
| 3: basal inferoseptal | 100 | 100 | 100 | 100 | 100 |
| 4: basal inferior | 80.00 | 100 | 100 | 93.75 | 95.00 |
| 5: basal inferolateral | 85.71 | 100 | 100 | 92.86 | 95.00 |
| 6: basal anterolateral | 100 | 100 | 100 | 100 | 100 |
| 7: medium anterior | 75.00 | 100 | 100 | 85.71 | 90.00 |
| 8: medium anteroseptal | 66.67 | 100 | 100 | 87.50 | 90.00 |
| 9: medium inferoseptal | 100 | 100 | 100 | 100 | 100 |
| 10: medium inferior | 71.43 | 100 | 100 | 86.67 | 90.00 |
| 11: medium inferolateral | 85.71 | 92.31 | 85.71 | 92.31 | 90.00 |
| 12: medium anterolateral | 83.33 | 100 | 100 | 93.33 | 95.00 |
| 13: apical anterior | 85.71 | 100 | 100 | 92.86 | 95.00 |
| 14: apical septal | 80.00 | 100 | 100 | 93.75 | 95.00 |
| 15: apical inferior | 85.71 | 100 | 100 | 92.86 | 95.00 |
| 16: apical lateral | 100 | 94.12 | 75.00 | 100 | 95.00 |
| global | 100 | 100 | 100 | 100 | 100 |

The analysis of the inter-observer variability highlighted that the DL-based segmentation compares well with both operators. The DSC, APD, and HD segmentation metrics showed a better agreement between the DL-based method and each individual observer than the agreement between the two observers, as shown in Figure 4a. As a result, the regional distribution of the T2* values exhibited small differences (<3 ms) associated with excellent intra-class correlation coefficients (ICC) in all the myocardial segments (inter-observer ICC), as displayed in Figure 4b.

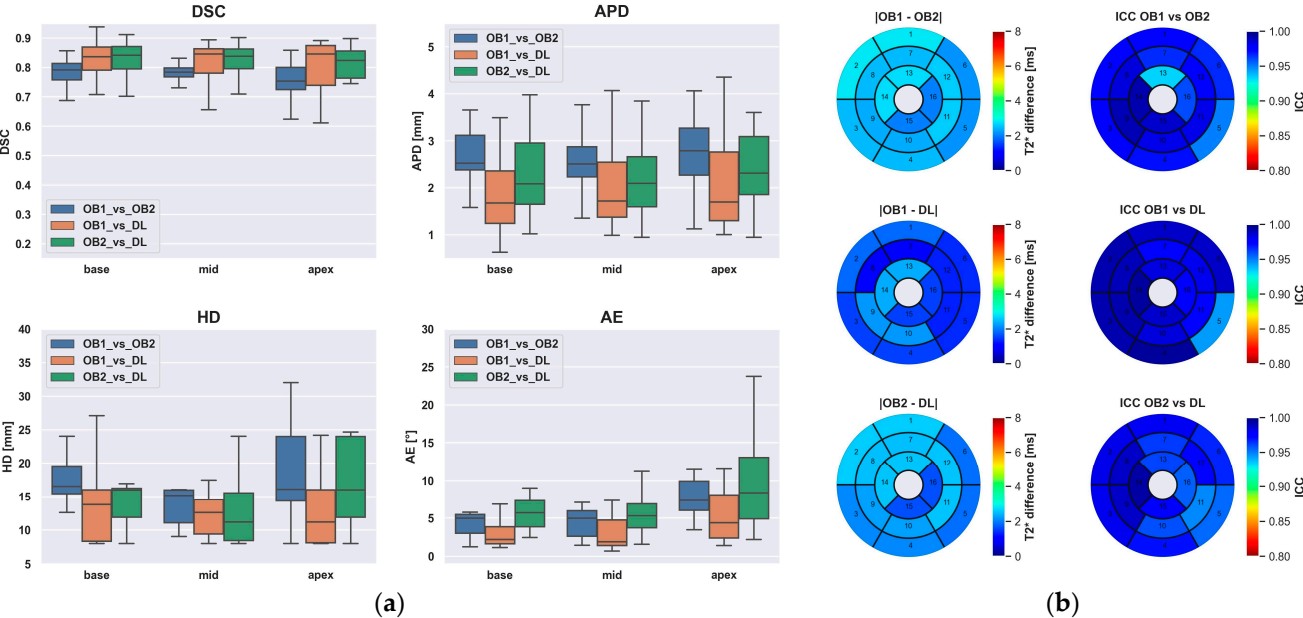

**Figure 4.** Inter-observer variability of the two experts compared to the fully-automated method. (**a**) Boxplot of the segmentation metrics. (**b**) Bull's-eye plots showing the regional distribution of segmental T2* differences according to the AHA model (left panels) and ICC between observers and the DL-based method (right panels).

## 4. Discussion

This paper presents a deep neural network-based workflow for automated myocardial segmentation and reporting of the AHA 16 sector model for pixel-wise T2* mapping. To the best of our knowledge, this is the first study on the application of deep learning methods for the global and regional quantification of T2* in the heart.

The CNN used in this study was based on the U-Net architecture and trained using a weighted loss function. A variety of network architectures and losses can be used for segmentation in medical imaging, and the best solution may vary for different applications. A comprehensive overview of these methods can be found in [34].

The proposed CNN was able to provide fully-automated segmentation of the multi-echo gradient-echo images with segmentation metrics in line with previous studies applied to other CMR acquisitions [35–37]. It is worth noting that, differently from cardiac cine analysis, for T2* quantification the endocardial and epicardial contours are usually traced in the myocardial midwall, i.e., with proper margins from the blood cavities, to avoid signal contamination from the blood. This might explain the lower Dice values (0.80–0.82) in the myocardial compared to some cardiac cine studies (0.87–0.94) [25,35], while our Dice coefficients in the LV cavity (0.90–0.93) well matched previous state-of-the-art techniques (0.90–0.94) [25,38]. Most interestingly, our inter-observer analysis showed a better agreement between the CNN segmentations and those provided by each individual observer compared to the agreement between the two experts, despite the fact that the second observer did not provide any training labels.

The key finding of our study is that global and regional T2* values assessed by the proposed DL method closely matched those obtained by expert CMR readers. The

Bland–Altman analysis demonstrated a 95% confidence interval for the global T2* of 3 ms compared to the manual measurements, which is more than acceptable for a fully-automated tool that does not require any user intervention. The agreement is worse for high T2* values since, due to technical constraints (maximum gradient-echo time), the T2* quantification loses accuracy and precision for cardiac T2* values longer than 20 ms [39]. Importantly, the developed DL method allowed us to achieve high diagnostic reliability in the identification of MIO (100% for global heart T2* values). The sensitivity was under 80% only for three myocardial segments, probably due to partial contamination of the blood pool in the myocardial segmentation of the DL method.

The inter-observer analysis revealed an excellent intraclass correlation between the DL method and each of the observers in all myocardial sectors of the AHA model (ICC range [0.944, 0.996]). This finding is a further demonstration that deep learning tools can reach the same accuracy as human experts with the advantage of eliminating the inter-observer variability since these models, once trained, produce purely deterministic outputs. This advantage can lead to improved precision in test-retest scans and, ultimately, can increase diagnostic and prognostic performance [40].

In terms of speed, our CNN can process three short-axis slices in less than one second, thus enabling a fully-automated T2* analysis that can be readily incorporated into the scanners immediately after image acquisition.

Nevertheless, our study has several limitations. First, all of the CMR images of this study were acquired with two MR scanners from a single vendor and with the same field strength (1.5T). Second, all T2* images are collected with the same bright-blood T2* mapping sequence. However, other sequences, such as the black-blood T2* technique, have been proposed [41]. Given the different contrast between the blood cavity and the myocardium, a re-training of the network or a transfer learning procedure [38] should be performed to adapt the present CNN to these types of images. Third, our CNN took as input all the ten multi-echo images of the gradient-echo acquisition. Alternative approaches could use as input a single image or even the T2* map directly.

## 5. Conclusions

We proposed a deep learning approach for the automatic assessment of global and regional T2*. The segmentation results showed a close agreement with manually annotated masks. No significant differences in the segmental T2* values were found compared to the manual measurements. This method could be effectively implemented in the clinical arena not only for faster, accurate, and quality-controlled analysis in trained centers but also for a fast implementation of new T2* CMR sites, in particular in regions of the world with economic constraints.

**Author Contributions:** N.M.: conceptualization, network training, data analysis, manuscript writing. A.M. (Antonella Meloni): conceptualization, formal analysis, writing—original draft preparation. V.P. and D.D.L.: conceptualization, methodology, software. P.K.: image analysis. L.P.: data curation. A.S., T.C., A.B. and A.M. (Antonella Massa): data collection. A.R.: methodology. F.C.: supervision. All authors assisted with interpretation, commented on drafts of the manuscript, and approved the final version. All authors have read and agreed to the published version of the manuscript.

**Funding:** The E-MIOT project receives "no-profit support" from industrial sponsorships (Chiesi Farmaceutici S.p.A. and Bayer). The funders had no role in study design, data collection and analysis, decision to publish, or preparation of the manuscript.

**Institutional Review Board Statement:** The study was conducted in accordance with the Declaration of Helsinki, and approved by the Institutional Ethics Committee of Area Vasta Nord Ovest (protocol code 56664, date of approval 8 October 2015).

**Informed Consent Statement:** Informed consent was obtained from all patients involved in the study.

**Data Availability Statement:** The data presented in this study are available on request from the corresponding author. The data are not publicly available due to privacy.

**Acknowledgments:** We would like to thank all the colleagues involved in the E-MIOT project (https://emiot.ftgm.it/) and all patients for their cooperation.

**Conflicts of Interest:** D.D.L. is presently an employee of Terarecon Inc.; his collaboration to the present study occurred before its present affiliation, his contribution to this article reflects entirely and only his own expertise on the matter, and he declares no competing financial interests related to the present article. All the other authors do not have competing interests to disclose concerning the present manuscript.

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
