# Peer review of "Fully Automated Regional Analysis of Myocardial T2* Values for Iron Quantification Using Deep Learning"

_electronics, doi:10.3390/electronics11172749_

Round 1

Reviewer 1 Report

Thank you for the opportunity to review this high quality manuscript. This novel application of a neural network to automatically quantify cardiac region-specific T2* maps is well described and valdiated in this methodological study. I am satisfied that the information presented is sound and of interest to readers. I have no suggested changes.

Author Response

We thank the Reviewer for his/her positive feedback.

Reviewer 2 Report

This study proposes automated segmentation of CMT T2* using deep learning approach. Extensive reliability and validity tests have been performed to demonstrate the robustness of the method. However, there are few points require improvements:

1) Section 2.3, line 139, "...previously validated, custom-written IDL-based soft- ware (Hippo-MIOT®) to obtain ground-truth data." Provide appropriate reference to support this.

2) Section 2.5, line 179, "...images was split into a training set (N=510 slices), validation set (N=60 slices), and test set (N=60 slices) for performance evaluation." It is unclear how the slices were obtained from 210 patients. It is also important to ensure that sets from the patients are mutually exclusive (e.g., patients used for training are not used in the testing set). Provide clear description about this.

3) Section 2.6.1. Provide acceptable range for Dice similarity coefficient with appropriate reference (e.g., >0.7 is deemed acceptable for medical image segmentation).

4) Section 2.6.2. "An ICC ≥ 0.75 was considered excellent, between 0.40 and 0.75 good, and < 0.40 unsatisfactory." Provide reference to support this classification.

5) It is suggested Section 2.6.3 to be renamed as "Assessment of Validity". Reliability is commonly used to refer to inter-/intra- grader variations. "Validity" is more appropriate to demonstrate the clinical application of the method. For examples,

van Netten, J. J., Clark, D., Lazzarini, P. A., Janda, M., & Reed, L. F. (2017). The validity and reliability of remote diabetic foot ulcer assessment using mobile phone images. Scientific reports, 7(1), 1-10.

Liew, G., Wang, J. J., Cheung, N., Zhang, Y. P., Hsu, W., Lee, M. L., ... & Wong, T. Y. (2008). The retinal vasculature as a fractal: methodology, reliability, and relationship to blood pressure. Ophthalmology, 115(11), 1951-1956.

6) Section 3.1, line 265 "...myocardial segments of the test data (320, 20x16)." It is unclear what does (320, 20x16) mean? 

7) Be consistent when citing the figure, for example, line 303 - 304, Figure 3a vs Figure 3(b)

8) Figure 3, please verify, is the unit in milliseconds, not millimeters?

9) Section 3.1, line 314. "The sensitivity was under 80% for three myocardial segments..." Give possible reasons of this in the Discussion section.

10) Section 4, line 346. "This might explain the lower Dice values in the myocardial compared to some cardiac cine studies [25,32], while our Dice coefficients in the LV cavity well matched previous state-of-the-art techniques [32,35]" Provide the range of the Dice values reported in the previous studies.

11) General comment: The similarity index of 36% is quite high, even after removing the references. Lower the similarity score to at least 24%.

Author Response

This study proposes automated segmentation of CMT T2* using deep learning approach. Extensive reliability and validity tests have been performed to demonstrate the robustness of the method. However, there are few points require improvements:

A: We would like to thank the Reviewer for the encouraging feedback and constructive critique and for the effort regarding this manuscript. We have addressed each of the raised concerns, which have substantially improved the manuscript.

1) Section 2.3, line 139, "...previously validated, custom-written IDL-based soft- ware (Hippo-MIOT®) to obtain ground-truth data." Provide appropriate reference to support this.

A: The appropriate reference has now been provided.

2) Section 2.5, line 179, "...images was split into a training set (N=510 slices), validation set (N=60 slices), and test set (N=60 slices) for performance evaluation." It is unclear how the slices were obtained from 210 patients. It is also important to ensure that sets from the patients are mutually exclusive (e.g., patients used for training are not used in the testing set). Provide clear description about this.

A: We used data from 170 patients for training, 20 for validation and 20 test phases. The patient groups were mutually exclusive, e.g. patients used for training were not used for validation or testing. Each patient had three slices, hence the total number of slices used for training was 510 (170x3), for validation was 60 (20x3) and for testing was 60 (20x3).

3) Section 2.6.1. Provide acceptable range for Dice similarity coefficient with appropriate reference (e.g., >0.7 is deemed acceptable for medical image segmentation).

A: As per Reviewer’s suggestion, the following sentence has now been added. “Following previous studies, a DSC > 0.7 was defined as substantial agreement, between 0.4 and 0.7 as moderate agreement, and  <0.4 as large variation (ref)”.

4) Section 2.6.2. "An ICC ≥ 0.75 was considered excellent, between 0.40 and 0.75 good, and < 0.40 unsatisfactory." Provide reference to support this classification.

A:  The appropriate reference has now been provided.

5) It is suggested Section 2.6.3 to be renamed as "Assessment of Validity". Reliability is commonly used to refer to inter-/intra- grader variations. "Validity" is more appropriate to demonstrate the clinical application of the method. For examples,

van Netten, J. J., Clark, D., Lazzarini, P. A., Janda, M., & Reed, L. F. (2017). The validity and reliability of remote diabetic foot ulcer assessment using mobile phone images. Scientific reports, 7(1), 1-10.

Liew, G., Wang, J. J., Cheung, N., Zhang, Y. P., Hsu, W., Lee, M. L., ... & Wong, T. Y. (2008). The retinal vasculature as a fractal: methodology, reliability, and relationship to blood pressure. Ophthalmology, 115(11), 1951-1956.

A: The Section 2.6.3 has been renamed as suggested.

6) Section 3.1, line 265 "...myocardial segments of the test data (320, 20x16)." It is unclear what does (320, 20x16) mean? 

A: We have now clarified that the total number of segments of the test data is 320: 20 patients x 16 segments.

7) Be consistent when citing the figure, for example, line 303 - 304, Figure 3a vs Figure 3(b)

A: The Figures are now cited always in the same way.

8) Figure 3, please verify, is the unit in milliseconds, not millimeters?

A: The unit is in milliseconds since it’s the unit of the T2* time constant.

9) Section 3.1, line 314. "The sensitivity was under 80% for three myocardial segments..." Give possible reasons of this in the Discussion section.

A: A possible reason of this result has been inserted in the discussion.

10) Section 4, line 346. "This might explain the lower Dice values in the myocardial compared to some cardiac cine studies [25,32], while our Dice coefficients in the LV cavity well matched previous state-of-the-art techniques [32,35]" Provide the range of the Dice values reported in the previous studies.

A: The range of the Dice values have been provided.

11) General comment: The similarity index of 36% is quite high, even after removing the references. Lower the similarity score to at least 24%.

A: We have now rephrased many sentences to reduce the similarity index.